# Air Plasma Functionalization of Electrospun Nanofibers for Skin Tissue Engineering

**DOI:** 10.3390/biomedicines10030617

**Published:** 2022-03-07

**Authors:** Abolfazl Mozaffari, Mazeyar Parvinzadeh Gashti

**Affiliations:** 1Department of Textile Engineering, Yazd Branch, Islamic Azad University, Yazd 8915813135, Iran; mozaffari@iauyazd.ac.ir; 2Research and Development Laboratory, PRE Labs Inc., 3302 Appaloosa Road, Unit 3, Kelowna, BC V1X 7Y5, Canada

**Keywords:** electrospinning, gelatin, nanofibers, plasma, skin tissue engineering

## Abstract

Nowadays, gelatin, a molecular derivative of collagen, has gained increasing interest in tissue engineering applications due to excellent biocompatibility, biodegradability, availability, process simplicity, and low costs. In this study, we fabricated tannic acid-crosslinked gelatin nanofibers by electrospinning method. In order to increase the bio-functionality of scaffolds, they were exposed to the atmospheric air plasma. Several analytical tools were used for evaluation of nanofibers including scanning electron microscopy (SEM), atomic force microscopy (AFM), attenuated total reflection Fourier transform infrared (ATR-FTIR) spectroscopy, X-ray diffraction (XRD), and water contact angle equipment (CA) together with biocompatibility study using fibroblast cells. Results demonstrated that atmospheric air plasma is not only able to improve the hydrophilicity of nanofibers but it also improves the bio-functionality against human skin fibroblast cells. Hence, we recommend atmospheric air plasma pre-treatment approach for the surface functionalization of gelatin nanofibers for skin tissue engineering applications.

## 1. Introduction

Gelatin is a natural biopolymer obtained from collagen under controlled hydrolysis. It has well-known properties for drug delivery, wound healing, and tissue regeneration applications [1,2,3,4,5,6,7]. Despite several advantages in biomedical engineering and regenerative medicine, gelatin has some major problems including melting at temperatures about or above 37 °C in water and hardening into gel at room temperatures. Therefore, specific pre-treatments are required for gelatin prior to applications in tissue engineering. Studies showed that gelatin can mimic different properties of collagen after the crosslinking approach with chemical reagents or physical processes [1]. In this context, aldehyde-based chemicals have been proposed by some researchers despite their moderate toxicity in tissue regeneration studies. Therefore, researchers have explored bioactive, non-toxic, and natural compounds to stabilize gelatin biomacromolecules [8]. Tannic acid is a plant polyphenol and a glucose derivative which is widely used as a natural reagent in the moisture, antioxidant, antimicrobial, antiviral, and anti-inflammatory products. Recent studies have approved that tannic acid can be effectively applied as a crosslinker for gelatin [9,10].

Electrospinning is a simple and effective method for producing long, uniform, and fine diameter nanofibers. Resultant scaffolds from different nanofibers have shown specific properties such as high surface-to-volume ratio, high porosity, and long lengths. Notably, they have been used in various products such as artificial tissues, biomedical products, electronics, sensors, and nanocomposites [11,12,13,14,15]. In the electrospinning process, different biomaterials can be utilized with effective mechanical properties in order to adapt and functionalize a biological environment [13,14,15]. On the other hand, surface modification of electrospun products has been an attractive method for increasing multifunctionality and biocompatibility properties [10]. A great number of studies demonstrated that different techniques can be applied for biofunctionalization of polymers including plasma, corona discharge, flame, photons, electron beam, ion beam, X-ray, and Gama-ray functionalization. However, plasma process of polymers has been known as a safe, cost-effective, and simple approach without affecting the bulk properties [16,17,18,19]. Our previous studies demonstrated that plasma process enhances the interactions between different polymers and increases the absorption properties of biomaterials. Moreover, protein molecules can easily be adsorbed on the plasma-treated membranes due to improvement in hydrophilicity [10,16]. This is owing to the fact that oxygen-containing molecules can be generated on plasma-treated biomaterials which is important in tissue engineering applications [20]. In continuation to our previous studies, the aim of our research is to study the effect of atmospheric air plasma method on bio-functionality of gelatin nanofibers against human skin fibroblast cells.

## 2. Materials and Methods

### 2.1. Electrospinning and Crosslinking Nanofibers

Gelatin powder (type A, Bio Reagent with code G1890 from porcine skin), tannic acid, and acetic acid (66%) were provided from Sigma Aldrich. According to the prior research, acetic acid was used as a solvent for gelatin electrospinning [9,10]. The spinning solution (20% *w/w*) was prepared by dissolving 2 g gelatin in 10 g acetic acid. Then, it was stirred at 30 °C for 4 h. The electrospinning process was conducted under the following conditions using the electrospinning device from Fanavaran nano-meghyas Co. (Tehran, Iran): high voltage value of 15 kV, distance of tip of needle to collector of 15 cm, and feeding rates of 0.6 mL/h. Then, nanofibers were collected onto an aluminum (Al) sheet. In order to generate crosslinked nanofibers, gelatin scaffolds were treated in a solution containing 5% *w/w* tannic acid and finally placed in a vacuum oven at 45 °C for 3 h. 

### 2.2. Plasma Functionalization

Crosslinked nanofiber scaffolds were treated in a PF-200 plasma DBD device (Nik Fanavaran Plasma Co. from Tehran, Iran) for 90 s in the presence of compressed atmospheric air. The voltage applied in this process was AC and up to 100 kV, the gas pressure was 2 L/min and the distance to nozzle was 3 mm. Plasma-treated samples were kept under vacuum prior to characterization.

### 2.3. Microscopic Evaluation

The morphology of gelatin nanofibers was illustrated with scanning electron microscope (SEM, LEO1455VP, ENGLAND). In this context, samples were first coated with sputter coating (Au layer) under vacuum conditions. We applied a pumping system along with a coating thickness controller of MTM-20 at a sputtering power of 30 W. The coating thickness and the target-to-substrate distance were 10 and 50 mm, respectively. The SEM working distance was set at 9 mm for all samples and a magnification of ×10 K (K = 1000) was used. SEM device was operated at a 25 kV accelerating voltage.

AFM was employed to evaluate the surface topography and the roughness of tissues on a non-contact mode (FemtoScan, Moscow, Russia). It had a single-beam Pt-coated fpN01S tip with a length of 130 µm, width of 35 µm, resonance frequency of 152 kHz, and force constant of 5.3 N/m. The root mean square (*RMS*) of nanofibers, as the most important parameter for roughness, was calculated according to Equation (1):(1)$$RMS=\sqrt{\frac{\sum_{n-1}^{N}\left(Zn-Zm\right)2}{N-1}},$$
where Zn is the height measurement of pixel *n* (from a total of *N* = 256 × 256 pixels), and Zm is the mean height.

### 2.4. Chemical Analysis

The ATR-FTIR spectra of nanofibers were assessed with the FTIR spectroscopy (Thermo Nicolet NEXUS 870 FTIR from Nicolet Instrument Corp., Portland, OR, USA). The spectra were collected after 64 scans per sample. 

### 2.5. Physical Properties and Wettability

X-ray diffraction studies of nanofibers were conducted from wide-angle X-ray diffractograms using a Philips X’Pert Pro Multipurpose X-ray Diffractometer operating at 40 mA. Ni-filtered Cu Ka radiation generated at 40 kV (k = 0.1542 nm) and the measured angle ranged from 4 to 70°, with the scan speed of 1°/min. 

Wettability of the scaffolds was assessed using a water contact angle system supported by a video camera equipment (Perkin Elmer Spectrum RX-1, Duluth, MN, USA). To achieve the contact angle (CA), the water droplet size was set at 0.5 mL and three samples were evaluated for each test; the average value of CA was finally reported.

### 2.6. Cell Culture and Microscopic Evaluation

We also studied the effect of atmospheric air plasma treatment of gelatin films on the human dermal fibroblast cell culture. HDF-1 (human dermal fibroblast) cells were purchased from Royan Institute, Iran. In this regard, primary cells, derived from human skin fibroblasts were cultured in Dulbecco’s modified Eagle’s medium (DMEM; Biosera, England) prepared with 10% fetal bovine serum (FBS; Gibco, Mississauga, ON, Canada), 100 IU/mL penicillin, and 100 μg/mL streptomycin. The cells were remained at 37 °C in 5% CO_2_ condition. Then, final cells (fourth passage) were cultivated on gelatin films with a size of 10 mm × 10 mm, by using a microscope slide cover glass (22 mm × 22 mm). After 24 h, films were washed twice with PBS, then fixed using 2.5% glutaraldehyde for 1 h at 4 °C, dehydrated by graded ethanol, and allowed to air-dry overnight. The dried films were finally imaged using SEM and optical microscopy with a cell seeding density of 10,000 cells/cm^2^ of medium. 

## 3. Results and Discussion

### 3.1. Microscopic Assessment

The surface morphology of gelatin nanofiber scaffolds was assessed before and after treatment with air pressure plasma, using SEM and AFM methods. Figure 1A,B indicates the results of SEM data for the untreated and plasma-treated gelatin nanofibers. There were no considerable changes on the surface, morphology, and the average diameter of nanofibers after plasma treatment. As a result of using acetic acid solvent, we previously found that several factors prevent bead-like patterns on electrospun nanofibers including electrostatic repulsion, surface tension, and viscoelastic properties. Therefore, we used optimized electrospinning conditions from our previous studies to generate smooth gelatin nanofibers [9,10]. In our latest study on argon and argon–oxygen plasma-treated gelatin nanofibers, SEM did not indicate any surface changes which is in agreement with the results from air pressure plasma-treated nanofibers. 

For an accurate surface topography evaluation of scaffolds, we led a complementary AFM analysis (Figure 1C,D). We also extracted the output results from AFM analysis, and roughness parameters were illustrated in Table 1. The surface of untreated fibers was smooth with an average RMS of 5.1 nm, as calculated from Figure 1C. The RMS for atmospheric-air plasma-treated nanofibers (Figure 1D) was 954.9 nm. A drastic increase in the surface roughness of nanofibers after plasma treatment can be resulted from several factors including the bombardment of energetic particles such as electrons, ions, radicals, neutrals, and excited atoms/molecules. On the other hands, chemical etching of polymer surfaces has been illustrated by plasma method due the bond breakage, chain scission, and chemical degradation [21,22]. In several research works, we established that the surface of polymer films and fibers can be oxidized after plasma functionalization.

### 3.2. Chemical Analysis

Figure 2 indicated ATR-FTIR spectra for the untreated and atmospheric-air plasma-treated gelatin nanofibers. We observed various bands arose for the untreated gelatin including a band at 3443 cm^−1^ due to N–H stretching of amide bond, C–H stretching at 2925 cm^−1^, C=O stretching at 1635–1651 cm^−1^, a band at 1449 cm^−1^ from C–C bond, and peaks at about 610-66 cm^−1^ related to C–H bond [23,24]. Several studies reported the amide I band in protein-based materials at 1650 cm^−1^ which was attributed to the random coil and alpha-helix conformations [25]. Notably, amide I band was associated with C-O stretching vibrations of peptide linkages in the backbone of proteins. However, amide II band was combined with N–H in plane bending, C–N stretching vibrations, and N–H out-of-plane wagging at 610 cm^−1^ [26].

For the atmospheric-air plasma-treated gelatin nanofibers, C–H stretching was observed at 3300 and 2947 cm^−1^. Moreover, a strong absorption band was exhibited at 3400–3500 cm^−1^ due to overlapping of N–H and O–H stretching vibrations. We also found the C=O stretching vibrations (amide I) at 1651 cm^−1^ [25]. In addition, a band at 1538 cm^−1^ was attributed to the C=O stretching (amide I) and N–H bending vibrations (amide II). The intensity of this band was changed after atmospheric-air plasma treatment. Furthermore, the intensity of bands at 1449 (C–C bond), 1241, and 1077 cm^−1^ (referred to C–O stretching bond) was also decreased after atmospheric-air plasma functionalization of gelatin nanofibers. Table 2 represented the effect of atmospheric-air plasma functionalization on the main peaks of gelatin.

### 3.3. X-ray Diffraction Analysis

The X-ray diffraction (XRD) test was conducted for the untreated and atmospheric-air plasma-treated gelatin nanofibers, and results are indicated in Figure 3. According to the spectrum for gelatin, the crystalline structure of fabricated scaffolds was demonstrated by peaks at 2θ = 38.5°, 45°, and 65.5°. Pena et al. established that the crystalline structure of gelatin generally originated from the triple helix conformation of protein macromolecules [27]. As can be seen in the XRD pattern of atmospheric-air plasma-treated gelatin, the location of crystalline peaks was not changed. However, the intensity of peaks was reduced in comparison with the untreated gelatin, which can be due to the oxidation or plasma chain session at gelatin crystalline regions [10]. 

### 3.4. Water Contact Angle Properties of Nanofibers

One of the most important properties of biomedical scaffolds is their hydrophilic/hydrophobic functionalities [28]. For this purpose, the contact angle (CA) of water droplets was measured on nanofibers using the device software. Table 3 represented the results of CA for the untreated and atmospheric-air plasma-treated gelatin nanofibers. As expected, gelatin indicated excellent wettability in comparison with different hydrophobic biopolymers including polylactic acid and poly caprolactone, with a water CA of 20.65. This is owing to its hydrophilic nature [29]. On the other hand, the water CA decreased to 11.8 for the atmospheric-air plasma-treated gelatin nanofibers due to the oxidation or attachment of polar groups [30]. Several studies postulated that the introduction of oxygen-polar domains on the nanofiber surfaces can be conducted by the atmospheric-air plasma method, which can subsequently lead to a drastic change in their hydrophilicity [28,29,30,31]. This observation is in good agreement with our ATR-FTIR and AFM results.

### 3.5. Effect of Atmospheric-Air Plasma on the Fibroblast Cell Culture

Fibroblast cells were put on the scaffolds and analyzed with inverted optical microscope along with SEM after 24 h. 

As it can be seen from the inverted microscope (Figure 4A with a magnification of 200×), fibroblast cells were flat. However, the microscopic examination did not reveal any major change in cell shape and adhesion to the atmospheric-air plasma-treated gelatin film (Figure 4B). According to Figure 4C from SEM, fibroblast cells were connected to the untreated gelatin film and their natural shapes were maintained after 24-h treatment with trypsin. On the other hand, there was no change in the shape of fibroblast cells on the plasma-treated gelatin film (Figure 4D). Interestingly, we found that the number of cells were increased on gelatin due to plasma functionalization. This result was in agreement with our recent study on argon and argon-oxygen plasma-treated gelatin nanofibers [10]. Several studies stated that plasma modification has a critical role in improving the hydrophilicity of scaffold surfaces which can further enhance the proliferation and differentiation of osteoblastic cells [32,33,34,35,36]. These researchers found that the number of oxygen atoms on tissue scaffolds was increased after atmospheric-air plasma treatment, thus the cell growth was improved.

## 4. Conclusions

We studied the effect of the atmospheric-air plasma on the physical, chemical, and bio-functional properties of gelatin nanofibers. Surface roughness of nanofibers was significantly increased after atmospheric-air plasma treatment due to ionization and chemical degradation. Furthermore, several chemical changes were proved by the ATR-FTIR spectroscopy. In contrast to our previous study on argon and argon–oxygen plasma-treated gelatin films, we did not observe any major changes in the XRD patterns of gelatin after atmospheric-air plasma treatment. However, the intensity of peaks was changed in comparison with the untreated gelatin. Furthermore, the water CA of gelatin was changed from 20.65° to 11.8° after atmospheric-air plasma treatment. This can be possibly due to the growth of new hydrophilic groups such as oxygen-containing polar domains on the surface of scaffolds. Notably, our process led to an improvement in the hydrophilicity, which was previously confirmed elsewhere [35,36]. Finally, the number of fibroblast cells was increased on the atmospheric-air plasma-treated gelatin. We conclude that plasma functionalization plays an important role in the skin tissue engineering.

## Figures and Tables

**Figure 1 biomedicines-10-00617-f001:**
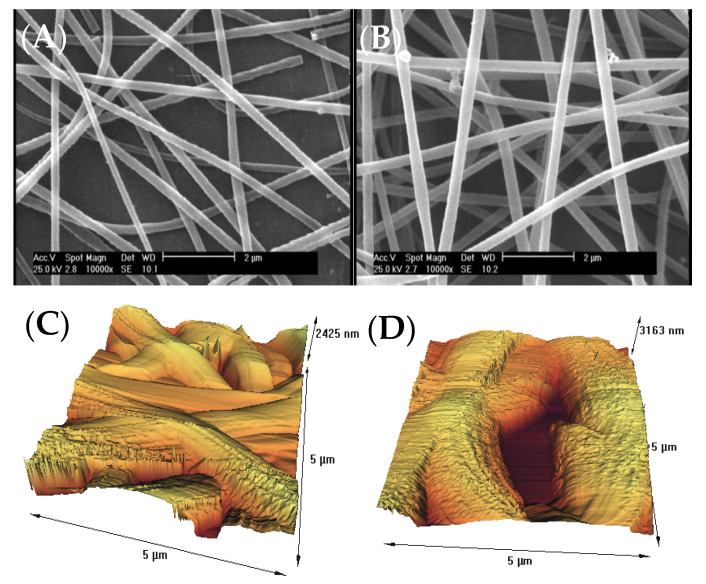
(**A**) SEM image of untreated gelatin nanofibers, (**B**) SEM image of atmospheric-air plasma-treated gelatin nanofibers, (**C**) AFM image of untreated gelatin nanofibers, (**D**) AFM image of atmospheric-air plasma-treated gelatin nanofibers.

**Figure 2 biomedicines-10-00617-f002:**
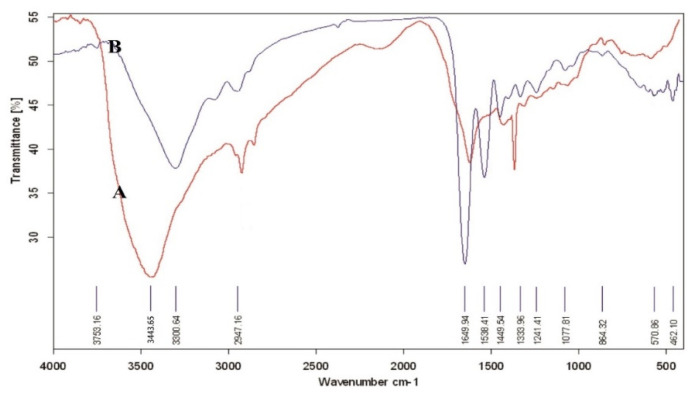
ATR-FTIR spectra of (**A**) untreated gelatin nanofibers, (**B**) atmospheric-air plasma-treated gelatin nanofibers.

**Figure 3 biomedicines-10-00617-f003:**
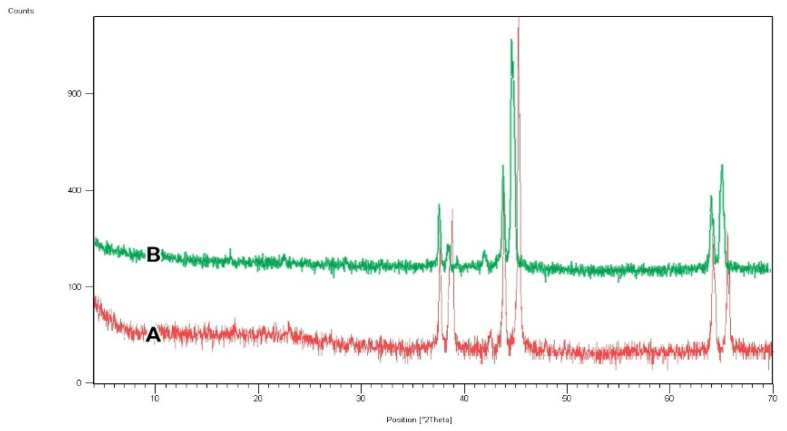
XRD spectra of (**A**) untreated, (**B**) atmospheric-air plasma-treated gelatin nanofibers.

**Figure 4 biomedicines-10-00617-f004:**
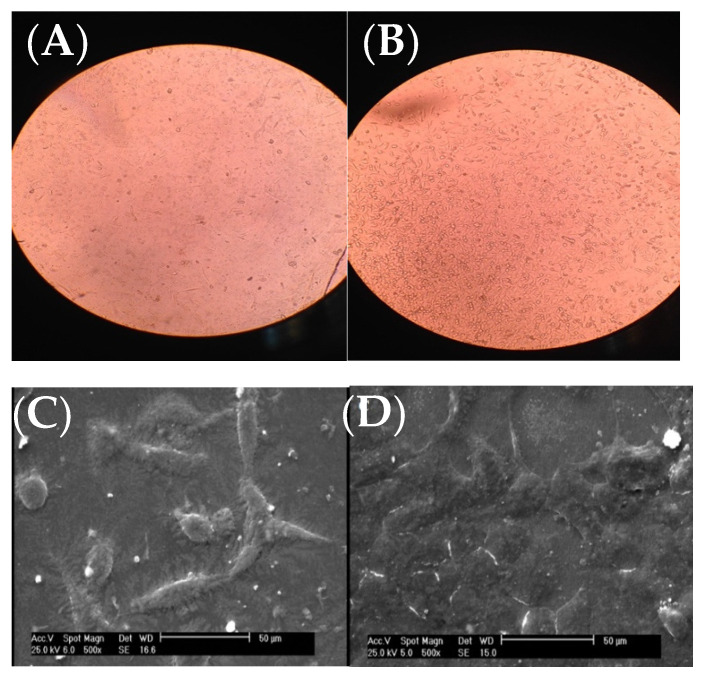
Images of fibroblast cells on gelatin films, (**A**) inverted optical microscope image from the untreated gelatin film, (**B**) inverted optical microscope image from the atmospheric-air plasma-treated gelatin film, (**C**) SEM image from the untreated gelatin film, (**D**) SEM image from the atmospheric-air plasma-treated gelatin film.

**Table 1 biomedicines-10-00617-t001:** Different parameters extracted from AFM analysis for the untreated and atmospheric-air plasma-treated gelatin nanofibers (standard deviations in parenthesis).

Samples	Plasma Exposure Time (s)	Maximum Peak Height, *Ra* (nm)	Maximum Valley Depth, *Rz* (nm)	Average Peak-to-Valley Height, *Rq* (nm)	*RMS* (nm)
Untreated nanofibers	0	6.5 (0.02)	276.8 (2.7)	47.4 (1.1)	5.1 (0.01)
Plasma-treated nanofibers	90	30.8 (0.1)	479.8 (5.3)	222.6 (2.5)	954.9 (4.3)

**Table 2 biomedicines-10-00617-t002:** ATR-FTIR band assignments of gelatin that were affected by atmospheric-air plasma treatment.

Peak Position (cm^−1^)	Band Assignment
610–669	–C–H
1333–1833	–CH_3_
1444–1449	C–C stretching
1635–1651	Amide I (C=O stretching)
2925	–CH stretching
3300	–CH stretching
3443	O–H stretching, NH stretching

**Table 3 biomedicines-10-00617-t003:** The average water CA (°) of (A) untreated, (B) atmospheric air plasma-treated gelatin nanofibers.

Samples	Water CA (°)	Image from Camera
A	20.6	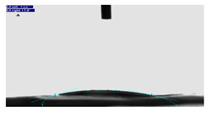
B	11.8	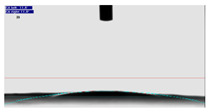

## Data Availability

The data used to support the findings of this study are included in the article. The data sets generated during and/or analyzed during the current study are available from the corresponding author on request.

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
