# Peer review of "Air Plasma Functionalization of Electrospun Nanofibers for Skin Tissue Engineering"

_biomedicines, 2022, doi:10.3390/biomedicines10030617_

Round 1

Reviewer 1 Report

This article deals with the surface functionalization of porous membranes obtained by electrospinning. The battery of tests carried out is quite standard that the results are not deepened. However, the results shown are of some interest. I propose to change or include the following:

Were samples kept under vacuum prior to characterization? It should be explained in materials and methods.

What tip did the authors use to perform the AFM tests? Include its characteristics (resonance frequency, elastic constant...).

Did the authors use positive and negative controls in cell culture evaluations? Explain.

Figure 1:

  • Can authors choose other SEM images to avoid significant differences between fiber diameters, fiber density...?
  • I would suggest homogenizing the point of view and the vertical scale of the 3D AFM images.

Table 1: include the standard deviations to the measurements and standardize the number of decimal digits.

Line 182: The abbreviation "XRD" was not entered before this point.

Line 196: The abbreviation “CA” was not entered before this point.

Line 199: “The droplet size was set to 0.5 ml. and three samples were evaluated for each test; finally, the average value of AC was reported”. This sentence should be changed to Materials and Methods.

Table 3: homogenize the number of decimal digits of the CA measurements.

Figure 4: Include IDs on SEM images (ie, "C" and "D").

Author Response

This article deals with the surface functionalization of porous membranes obtained by electrospinning. The battery of tests carried out is quite standard that the results are not deepened. However, the results shown are of some interest. I propose to change or include the following:

+++ We thank our esteemed reviewer for encouraging feedback.

Were samples kept under vacuum prior to characterization? It should be explained in materials and methods.

+++ As recommended by our esteemed reviewer, we added the statement that samples were maintained in a vacuum chamber prior to characterization.

What tip did the authors use to perform the AFM tests? Include its characteristics (resonance frequency, elastic constant...).

+++ As recommended by our esteemed reviewer, we added more details about the equipment at the Materials and Methods section.

Did the authors use positive and negative controls in cell culture evaluations? Explain.

+++ We thank our esteemed reviewer for this question. In fact, we only used one type of cell evaluation on nanofibers. In this research the connection and the shape of cells were analyzed and compared on untreated and plasma treated samples. 

Figure 1:

  • Can authors choose other SEM images to avoid significant differences between fiber diameters, fiber density...?

+++ As recommended by our esteemed reviewer, we changed both SEM images to avoid the differences in diameter and density properties.

  • I would suggest homogenizing the point of view and the vertical scale of the 3D AFM images.

+++ As recommended by our esteemed reviewer, we replaced AFM images from different point of view.

Table 1: include the standard deviations to the measurements and standardize the number of decimal digits.

+++ As recommended by our esteemed reviewer, we added the SD to the table and standardized the decimal numbers.

Line 182: The abbreviation "XRD" was not entered before this point.

+++ As recommended by our esteemed reviewer, it was corrected.

Line 196: The abbreviation “CA” was not entered before this point.

+++ As recommended by our esteemed reviewer, it was corrected.

Line 199: “The droplet size was set to 0.5 ml. and three samples were evaluated for each test; finally, the average value of AC was reported”. This sentence should be changed to Materials and Methods.

+++ As recommended by our esteemed reviewer, we moved this sentence to Materials and Methods.

Table 3: homogenize the number of decimal digits of the CA measurements.

+++ As recommended by our esteemed reviewer, it was corrected.

Figure 4: Include IDs on SEM images (ie, "C" and "D").

+++ As recommended by our esteemed reviewer, it was corrected.

Reviewer 2 Report

We recommend minor revisions as follows:

- Complete the summary by introducing on page 1, line 17, the following sentence: "...angle (CA) equipment together with biological biocompatibility on fibroblast cells.";

- Figure 4 – erase small letters (c) and (d); keep only the capital letters (C and D);

- page 7, line 232 – replace Figure 4d with Figure 4D;

- page 1, line 9: Correspondence:

- renumbering subchapters of Chapter 2. Materials and Methods (2.1, 2.2, etc., instead of the already reviewed 3.1, 3.2, etc.)

- References:     - bold figures for years (R3 – 2012; R22 – 2002);

                            - non-italic figures for the year (R25 – 2007).

Author Response

We recommend minor revisions as follows:

+++ We thank our esteemed reviewer for encouraging comment.

- Complete the summary by introducing on page 1, line 17, the following sentence: "...angle (CA) equipment together with biological biocompatibility on fibroblast cells.";

+++ As recommended by our esteemed reviewer, it was added in the abstract section.

- Figure 4 – erase small letters (c) and (d); keep only the capital letters (C and D);

+++ As recommended by our esteemed reviewer, it was corrected.

- page 7, line 232 – replace Figure 4d with Figure 4D;

+++ As recommended by our esteemed reviewer, it was corrected.

- page 1, line 9: Correspondence:

+++ As recommended by our esteemed reviewer, it was corrected.

- renumbering subchapters of Chapter 2. Materials and Methods (2.1, 2.2, etc., instead of the already reviewed 3.1, 3.2, etc.)

+++ As recommended by our esteemed reviewer, it was corrected.

- References:     - bold figures for years (R3 – 2012; R22 – 2002);

                            - non-italic figures for the year (R25 – 2007).

+++ As recommended by our esteemed reviewer, it was corrected.

Reviewer 3 Report

In this manuscript, “Air Plasma Functionalization of Electrospun Nanofibers for Skin Tissue Engineering” by Mozaffari and Gashti reported the fabrication and characterization of tannic acid-crosslinked gelatin nanofibers by electrospinning method. Although some preliminary results were demonstrated, this work is premature for publication. Therefore, I would suggest some other biomaterials related journals might be more suitable of this manuscript. Here are the comments and suggestions:

  1. The content of this manuscript is more related with biomaterials than biomedicine; therefore, some other journals might be more suitable.
  2. Only some OM or SEM pictures with fibroblast cells were demonstrated, without and in vivo or animal models.

Author Response

In this manuscript, “Air Plasma Functionalization of Electrospun Nanofibers for Skin Tissue Engineering” by Mozaffari and Gashti reported the fabrication and characterization of tannic acid-crosslinked gelatin nanofibers by electrospinning method. Although some preliminary results were demonstrated, this work is premature for publication. Therefore, I would suggest some other biomaterials related journals might be more suitable of this manuscript. Here are the comments and suggestions:

+++ We thank our esteemed reviewer for the comments. In fact, we submitted this article to the Special Issue: Biomaterial Modifications and Improvement of Their Biocompatibility.

It can be reached through the following link:

https://www.mdpi.com/journal/biomedicines/special_issues/bio_compat

As can be seen by our esteemed reviewer, our article completely matches with the scope and key words of this special issue.

  1. The content of this manuscript is more related with biomaterials than biomedicine; therefore, some other journals might be more suitable.

+++ We thank our esteemed reviewer for the comments. Please see our above statements.

  1. Only some OM or SEM pictures with fibroblast cells were demonstrated, without and in vivo or animal models.

+++ We thank our esteemed reviewer for this comment. In fact, we did not have access to the required labs for conduct in-vivo studies due to the impact of pandemic on the availability of research labs. In the next step, authors of this article have planned to conduct additional in-vivo experiments and prepare the next article for publication after having access to the authorized labs.
